# The quaternary architecture of RARβ–RXRα heterodimer facilitates domain–domain signal transmission

Vikas Chandra[1], Dalei Wu[1,2], Sheng Li[3], Nalini Potluri[1], Youngchang Kim[4] & Fraydoon Rastinejad[1]

Assessing the physical connections and allosteric communications in multi-domain nuclear receptor (NR) polypeptides has remained challenging, with few crystal structures available to show their overall structural organizations. Here we report the quaternary architecture of multi-domain retinoic acid receptor β–retinoic X receptor α (RARβ–RXRα) heterodimer bound to DNA, ligands and coactivator peptides, examined through crystallographic, hydrogen–deuterium exchange mass spectrometry, mutagenesis and functional studies. The RARβ ligand-binding domain (LBD) and DNA-binding domain (DBD) are physically connected to foster allosteric signal transmission between them. Direct comparisons among all the multi-domain NRs studied crystallographically to date show significant variations within their quaternary architectures, rather than a common architecture adhering to strict rules. RXR remains flexible and adaptive by maintaining loosely organized domains, while its hetero-dimerization partners use a surface patch on their LBDs to form domain-domain interactions with DBDs.

[1] Integrative Metabolism Program, Sanford Burnham Prebys Medical Discovery Institute, Orlando, FL 32827, USA. [2] Shandong University-Helmholtz Institute of Biotechnology, State Key Laboratory of Microbial Technology, School of Life Sciences, Shandong University, Qingdao, Shandong 266237, China. [3] Department of Medicine and UCSD DXMS Proteomics Resource, University of California, San Diego, La Jolla, CA 92023, USA. [4] Structural Biology Center, Biosciences Division, Argonne National Laboratory, Argonne, IL 60439, USA. Vikas Chandra and Dalei Wu contributed equally to this work. Correspondence and requests for materials should be addressed to F.R. (email: frastinejad@sbpdiscovery.org)

Nuclear receptors (NRs) are a family of transcription factors that respond to lipophilic ligands and control a variety of metazoan gene programs[1, 2]. Their polypeptides consist of a variable N-terminal domain (NTD), a central DNA-binding domain (DBD), and a 12-helical ligand-binding domain (LBD) located at their C-terminus[3]. Physical connections and the routes for allosteric communications between domains have remained difficult to assess, given the relatively few examples of crystal structures involving multi-domain NR complexes. Employing full-length or multi-domain NR complexes bound to DNA, crystal structures have been reported to date only for the peroxisome proliferator-activated receptor γ–retinoic X receptor α (PPARγ–RXRα) heterodimer (PDB codes: 3DZY, 3DZU, and 3E00)[4], the hepatocyte nuclear factor 4α (HNF-4α) homodimer (PDB code: 4IQR)[5], and the liver X receptor β (LXRβ)–RXRα heterodimer (PDB code: 4NQA)[6]. In some cases, multi-domain NRs have been studied by alternate biophysical techniques and those data were interpreted to suggest possible structural models. For example, RXRα was studied heterodimerized with RARα, PPARγ2 and vitamin D nuclear receptor (VDR) using small-angel X-ray scattering, fluorescent studies and/or small-angle neutron scattering[7]. These interpretations were further used to suggest the existence of a common architecture of NR heterodimers on DNA direct repeats[7]. However, the crystallographic models have not been supportive of these structural models, and have been showing instead that the quaternary structures of RXR heterodimers consist of significant variations without a commonly adopted organization[8–10]. To understand larger NR complexes with full-length coactivators, the quaternary structure of an active complex of DNA-bound estrogen receptor α (ERα), steroid receptor coactivator 3 (SRC-3/NCOA3), and a secondary coactivator (p300/EP300) has been described based on cryoelectron microscopy (cryo-EM)[11].

Retinoids bind to a subclass of NRs in mammals consisting of three retinoic acid receptors (RARα/β/γ) and three retinoid X receptors (RXRα/β/γ). Through heterodimerization between an RAR and an RXR, productive and functional transcription factors are formed within this subclass of NRs. Ligand binding modulates the transcriptional activities of these receptor heterodimers by altering the protein surface conformations at each of their LBDs and shaping preferences for coactivators and corepressors. Specifically, ligand-binding repositions helix-12 of the LBDs into an active conformation that fosters the binding of LXXLL motifs of coactivators, including those belonging to members of the steroid receptor coactivator (SRC or p160) family. The repertoire of retinoid actions through these NRs includes a variety of essential biological processes such as embryogenesis, organogenesis, cell growth, differentiation, and apoptosis[12]. The six mammalian NRs for retinoids are encoded by distinct genes, but they share close structural homology at the level of their LBDs and DBDs[13].

The RAR and RXR proteins are among the most intensively studied for their structural properties, but most structural characterizations to date have focused on isolated LBD domains, and have not been successful with their heterodimeric multi-domain polypeptide complexes[13, 14]. The retinoic acid receptors exhibit high-affinity binding to endogenous molecules that include all-trans retinoic acid (REA, for RARs) and 9-cis retinoic acid (9CR, for RXRs). Ligand binding to RARs leads to modulation of gene programs that are particularly relevant in embryonic development and pattern formation[13, 15]. In adult organisms, RAR ligands can directly impact cellular differentiation, and synthetic ligands that bind and regulate RXRs and RARs have been developed as therapeutic agents for cancers (promyelocytic leukemia, myelodysplastic syndrome, cutaneous T-cell lymphoma, and squamous carcinoma of the skin) and inflammatory diseases (severe acne and psoriasis)[13, 16].

To directly visualize how different domains of RAR and RXR polypeptides physically interact within and between their polypeptides, we conducted crystallographic studies using multi-domain recombinant RARβ and RXRα proteins. Both receptors' ligands, including REA for RARβ and 9CR for RXRα, their idealized DNA response element (RARE) and coactivator peptides were included to visualize how all these components are configured and interact within the overall complex. We further utilized hydrogen–deuterium exchange mass spectrometry (H/D-ex MS) to determine if signals at one domain can register their effects across the heterodimer to distal domains, and further relied on functional transcriptional studies and mutagenesis to validate our understanding of domain–domain junctions. Finally, we sought here to understand the quaternary architecture of RARβ–RXRα heterodimer in the wider context of other recently reported crystal structures of multi-domain NR complexes, to assess their level of overall variations and to pinpoint specific sites where domain–domain junctions are formed in each case.

## Results

**Overall structure of RARβ–RXRα heterodimer.** We set out to obtain the quaternary architecture of multi-domain RARβ–RXRα heterodimer including its domain–domain couplings, using X-ray crystallography. To do this, we biochemically prepared the human RARβ–RXRα protein complex (involving their DBD, hinge, and LBD regions, as shown in Fig. 1a) by co-expression of two proteins in *Escherichia coli*. Using extensive crystallization screens that utilized various sized DBD–hinge–LBD segments of each isotype of RXRs and RARs, and a variety of known RAR ligands, we successfully obtained crystals of only the RARβ–RXRα heterodimer on a DR1 DNA in the presence of REA and 9CR retinoic acids, and with LXXLL synthetic peptides derived from SRC-2's second (middle) NR box. The crystal structure was solved at 3.5 Å resolution by molecular replacement using known RAR–RXR DBD and LBD dimeric structures (PDB codes: 1DSZ[17] and 1XDK[18]) as the search models (Table 1), leading to a readily interpretable electron density map that distinctly revealed the overall quaternary architecture, and the locations of bound DNA, ligands (REA was bound inside RARβ LBD and 9CR was bound inside RXRα LBD) and coactivator peptides (Supplementary Fig. 1).

Figure 1b shows the arrangement of two subunits and the mode by which the RARβ–RXRα heterodimer binds to DR1 DNA, ligands, and coactivators. The DBD and LBD of RARβ are physically connected; however, the corresponding two domains of RXRα are spatially displaced from each other without any physical contacts between them, and each of these domains locates on the opposite side of the double-strand DNA (Fig. 1b). Furthermore, we could not observe ordered electron density for the hinge region connecting those two RXRα domains, suggesting this region remains flexible and disordered. A series of inter-subunit domain–domain interactions are clearly formed between RARβ and RXRα, involving LBD–LBD and DBD–DBD interfaces (Supplementary Fig. 2a), with the latter forming directly atop the DR1 DNA, as we previously described[17]. Importantly, the structures of individual LBD and DBD domains are essentially identical to those previously reported[17, 18] (the Cα rmsd values are 0.4838 and 0.5143 for RAR LBD and DBD, 0.6515 and 0.4477 for RXR LBD and DBD, respectively; Supplementary Fig. 2b), as are the modes of LBD's specific interactions with retinoic acid ligands, coactivator LXXLL peptides, and the DBDs' interactions with the DNA half-sites. These observations together indicate that no domain requires major internal distortions in adopting the quaternary state of the multi-domain heterodimer.

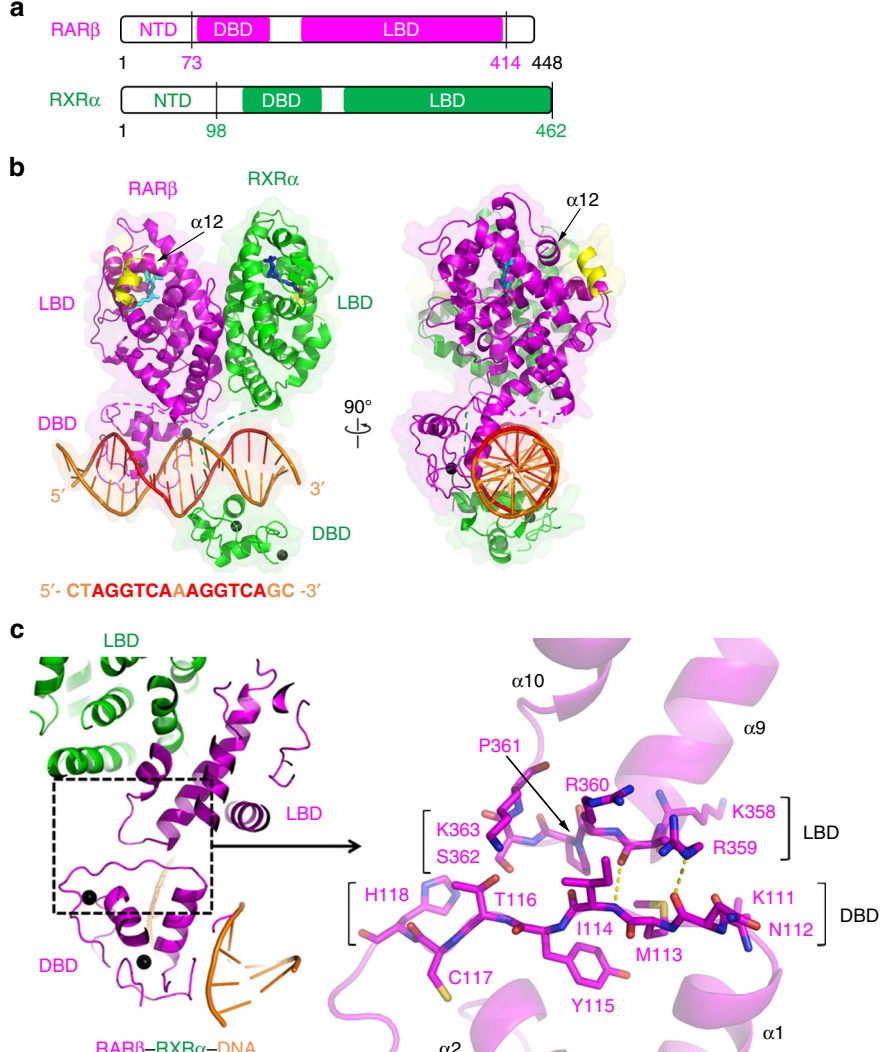

**Fig. 1** Overall structure of RARβ–RXRα–DNA complex and the DBD–LBD interface of RARβ. **a** Schematic representation shows the domain arrangements of RARβ and RXRα, with the residue numbers labeled for the protein constructs used in crystallization. **b** Overall structure of the RARβ–RXRα heterodimer on DNA in two views. The disordered hinge regions of RARβ (residues 153–173) and RXRα (residues 210–225) are shown in dashed lines. The colors used for RARβ, RXRα, DNA, coactivator peptides, REA, 9CR, and Zn are magenta, green, orange, yellow, cyan, blue, and black, respectively. The AGGTCA sites within DR1 DNA sequence are colored in red. The helix-12 (α12) of RARβ LBD is also labeled by black arrows. **c** The DBD–LBD interface within the RARβ protein, together with an enlarged view on the right side showing the participating residues. The two hydrogen bonds (formed by N112 [O] and R359 [NE], and by I114 [N] and R359 [O]) at the interface are shown as yellow dashed lines

Both RXRα and RARβ assume the active conformation at their LBDs. This conformation is defined by both receptors having their helix-12 appropriately positioned by ligands to facilitate the recruitment of coactivator LXXLL motifs (Fig. 1b). Importantly, two coactivator peptides are observed to be stably and equivalently bound to the heterodimer, with one peptide bound at each LBD, contradicting a previous study, based on artificially formed RAR LBD homodimers, that suggested the binding of one LXXLL peptide to an LBD would disfavor and prevent the binding of another peptide to the second LBD of RAR–RXR heterodimers through allosteric effects between LBDs[19]. We have previously pointed out that proposed mechanism for allostery was flawed for a number of reasons, including the reliance on RARβ homodimers, a species not observed in cells or known to be physiologically relevant[8]. Moreover, the previous structural analyses of RAR–RXR LBD–LBD heterodimer clearly showed simultaneous binding of a peptide to each receptor LBD[18], consistent with our current observations of two coactivator peptides binding to the multi-domain RARβ–RXRα heterodimer.

The binding of LXXLL motifs equivalently to both subunits of the RARβ–RXRα heterodimer has also been observed in other heterodimers, including to multi-domain PPARγ–RXRα, LXRβ–RXRα, and HNF-4α dimeric complexes[4–6].

**The DBD–LBD interface of RARβ allows allosteric transmission.** The disordered hinge region and lack of domain–domain interactions within RXRα polypeptide are consistent with its required flexibility in adapting to its many dimerization partners among the NR family[4]. But in the case of RARβ, a well-formed interface is observed to physically connect the DBD and LBD segments of the polypeptide (Fig. 1c). This interface is mainly established via the DBD's loop immediately following α1 helix, and the LBD's loop between its α9 and α10 helices (Supplementary Fig. 3). In addition to two hydrogen bonds formed between R359 and main-chain atoms of N112 and I114 (Fig. 1c), hydrophobic contacts also contribute to the interactions at this interface. The buried surface areas of DBD and LBD are 345 and

**Table 1 Data collection and refinement statistics**

|  | RARβ–RXRα–DNA complex |
|---|---|
| *Data collection* |  |
| Space group | P 21 |
| Cell dimensions |  |
| *a, b, c* (Å) | 52.47, 77.40, 112.11 |
| *α, β, γ* (°) | 90.0, 90.37, 90.0 |
| Resolution (Å) | 50.0-3.50 (3.56–3.50)[a] |
| $R_{sym}$ or $R_{merge}$ | 11.0 (43.1) |
| $I/\sigma I$ | 12.6 (1.8) |
| Completeness (%) | 84.0 (52.2) |
| Redundancy | 4.1 (2.5) |
|  |  |
| *Refinement* |  |
| Resolution (Å) | 47.4-3.51 (3.63–3.51) |
| No. reflections | 13,381 (348) |
| $R_{work}/R_{free}$ | 22.0/26.8 (30.8/35.2) |
| No. atoms |  |
| Protein/DNA | 5512 |
| Ligand/ion | 48 |
| Water | 0 |
| B-factors |  |
| Protein/DNA | 61.71 |
| Ligand/ion | 51.94 |
| Water | – |
| R.m.s deviations |  |
| Bond lengths (Å) | 0.003 |
| Bond angles (°) | 0.741 |

[a]Highest resolution shell is shown in parenthesis

312 Å², respectively, as calculated by *PISA*[20]. Protein sequence alignment of RARα, RARβ, and RARγ shows residues involved in this DBD–LBD interface are highly conserved within all three isoforms of RARs (Supplementary Fig. 4).

Using distinct classes of RARβ ligands, we asked if switching from the agonist REA to an antagonist BMS-189453[21] produced effects outside the RARβ LBD (Fig. 2). H/D-ex MS studies were used to readily and accurately address this question, given the lack of success in crystallizing alternate complexes with different ligands or DNA. The deuteration level (indicated with the rainbow colors in Fig. 2a) of residues monitored by MS, correlates well with the flexibility of local structures. This technique can also identify conformational changes induced by different ligands or DNA when used in a comparative manner (i.e., subtraction of the deuteration levels of each residues in different conditions as shown with the blue-white-red heat maps in Fig. 2a). H/D-ex MS showed that helix-12, associated with the activation function-2, whose conformational and dynamic state is known to be highly responsive to ligand binding to RAR[8], registered clear changes (increased deuteration level) when ligands were switched from agonist REA to antagonist BMS-189453 (Fig. 2a, b), indicating less stable conformation of this region due to the switch. Additionally, the DBD and hinge region of RARβ registered an altered H/D-ex MS pattern (increased deuteration level) when the DNA response element was switched from DR1 to DR5 (Fig. 2a), indicating these regions undergo conformational changes upon binding of DNAs with different spacers.

Given the observed sensitivity in the H/D-ex MS studies noted above, we further applied this method to ask if the DBD–LBD interactions within the RARβ protein could signal the change of ligands inside the LBD allosterically to the DBD through their physical domain–domain connection. Previously we had shown evidence for allosteric transmission from the LBD to the DBD in the HNF-4α homodimer by using mutations introduced within the LBD[5]. But because the ligand bound to HNF-4α was not exchangeable, we could not assess if switching ligands at LBD would be

registered distally at the DBD. The H/D-ex MS studies conducted here on RARβ–RXRα, now point us to three sites on the RARβ DBD (positioned directly before and after helix α1, and at the C-terminal end of the DBD) that show altered H/D-ex MS patterns when we switched ligands between REA and BMS-189453 (Fig. 2a, c).

To further assess how ligands may allosterically alter the properties of the RARβ–RXRα heterodimer, we used a wider group of agonists and antagonists and tested their effects on the RARβ–RXRα affinity for DR1 DNA (Fig. 2d). Our biochemical DNA-binding assay was based on detecting fluorescence polarization of 5′-FAM labeled DNA in the presence of increasing concentrations of the heterodimer protein to obtain a $K_D$ value for DNA affinity. The results showed that distinct RAR ligands lead to variations in DNA-binding affinities of the heterodimer within a $K_D$ range of about 10–70 nM. The affinities for DNA binding were lower with agonists ($K_D$ of 27–68 nM) as compared to antagonists ($K_D$ of 12–16 nM) (Fig. 2d). A possible explanation is suggested by the H/D-ex MS data, where the DNA-binding region is somewhat more flexible and prone to H/D exchange with an agonist (REA) than that with an antagonist (BMS-189453) (Fig. 2a, c).

**The functional importance of the DBD–LBD interface**. Using cell-based functional studies, we next tested the importance of the RARβ DBD–LBD interface for the transcriptional activities of RARβ–RXRα from different response elements. Reporter assays were used that incorporated response elements derived from *ANGPTL4* (DR1) and *CYP26A1* (DR5), both of which are target genes of RARβ–RXRα (Fig. 3a). Transfection of wild-type RARβ alone decreased DR1 reporter activity, and the level of repression was enhanced further when REA was added. On DR5, transcriptional activity was increased when transfecting in wild-type RARβ, and that activity was further enhanced by REA. These findings are consistent with previous reports indicating that DR1 and DR5 elements allow transcriptional repression and activation, respectively, by RAR–RXR heterodimers[22].

We then directly probed the importance of RARβ DBD–LBD connectivity for functional activity, by introducing mutations at this domain–domain junction and examining effects on transcriptional regulation from response elements. Point mutations introduced at RARβ DBD's DNA-binding helix (E99A and R106A) abolished the transcriptional repression seen with wild-type RARβ on DR1, as expected (Fig. 3a). Then examining the specific mutations positioned within the RARβ DBD–LBD interface, we further found loss of repression when compared to wild-type RARβ, suggesting that this interface is important for the transcriptional repression of RARβ–RXRα from the DR1 element.

Interestingly, these same mutations positioned specifically at the DBD–LBD interface of RARβ also compromised the transcriptional activation from the DR5 reporter (Fig. 3a). This finding suggests that the same RARβ DBD–LBD interface may be similarly critical in establishing the binding of the RARβ–RXRα heterodimer on DR5. While the integrity of this particular intramolecular interface within the RAR polypeptide appears simultaneously important for both DR1 and DR5 complexes, the intermolecular interfaces between RXR and RAR should prove substantially different, since the binding polarity of the RXR and RAR subunits on DR1 vs. DR5 half-sites is known to be reversed[22].

Finally, we directly tested the influence of these mutations positioned at the intramolecular domain—domain interface of RARβ on the heterodimer's overall affinity for DNA. To do this, we co-purified mutant RARβ proteins where residues at the DBD–LBD interface were altered along with WT RXRα. We again used fluorescence polarization studies to obtain the affinities of the resulting heterodimers for both DR1 and DR5 DNAs, in the

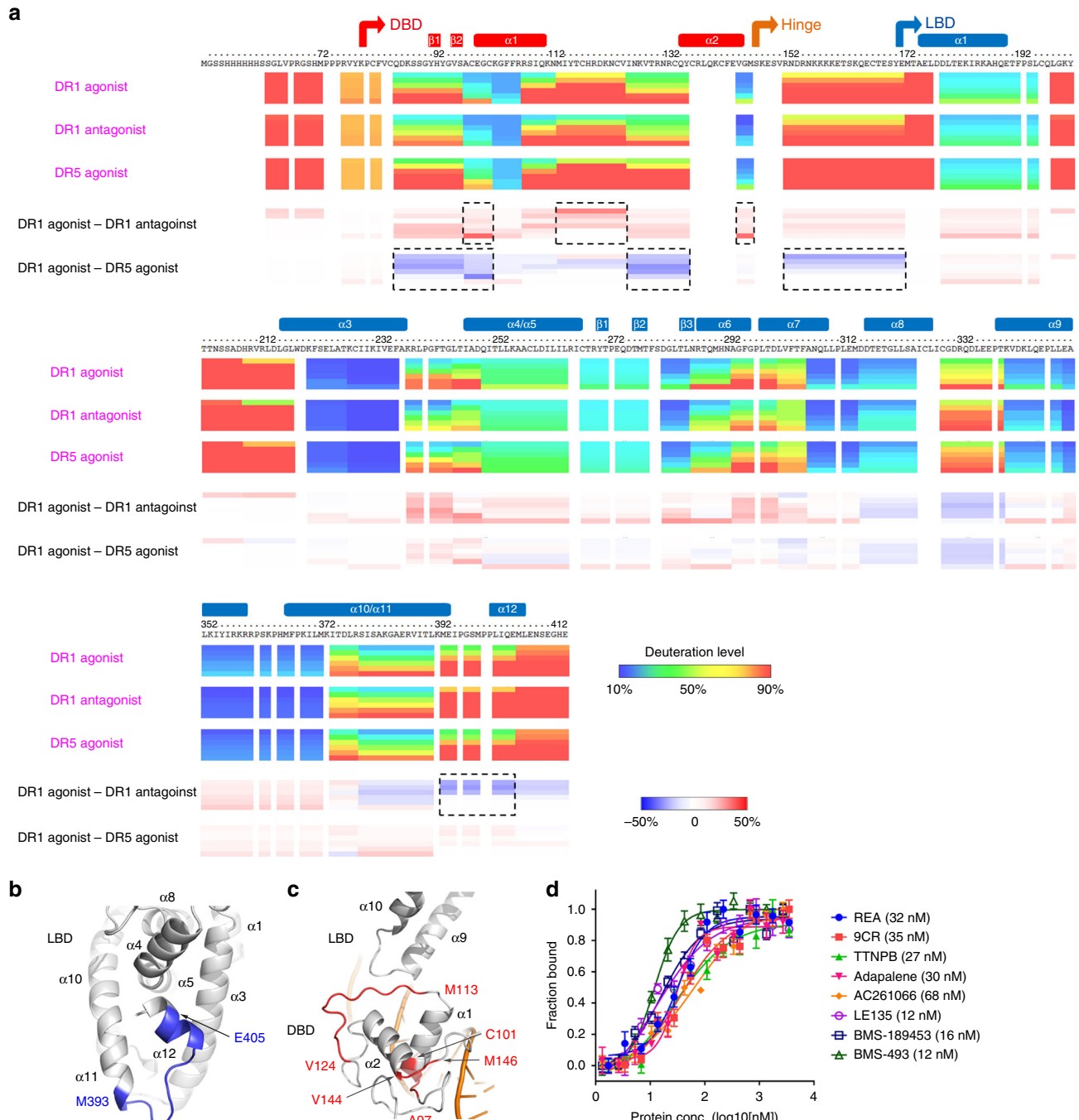

**Fig. 2** Dynamic features of RARβ protein in the RARβ–RXRα heterodimer measured by H/D-ex MS and effects of different ligands on DNA binding. **a** The three ribbon maps showing the deuteration level of RARβ residues are from states in the presence of DR1 DNA and agonist REA (1st row), DR1 DNA and antagonist BMS-189453 (2nd row), DR5 DNA and agonist REA (3rd row), respectively. The 4th row shows the differences between 1st and 2nd rows (i.e., DR1 agonist minus DR1 antagonist), while the 5th shows those between 1st and 3rd (i.e., DR1 agonist minus DR5 agonist). The regions with relative larger differences are boxed by dotted lines. For each map, there were six time points (30, 100, 300, 1000, 3000, and 10,000 s, from top to bottom). The domain ranges and secondary structures are labeled above the protein sequence. **b, c** Regions in RARβ LBD (**b**) and DBD (**c**) displaying more (red) or less (blue) dynamics in the hydrogen/deuterium exchange, when comparing the data from DR1 and agonist (REA) to those from DR1 and antagonist (BMS-189453) (i.e., corresponding to the 4th row in Fig. 2a). The secondary structures and residue ranges are labeled accordingly. **d** DNA-binding affinities of RARβ–RXRα heterodimer to DR1 DNA measured in the presence of different ligands. REA, 9CR, TTNPB, adapalene, and AC261066 are agonists for RARβ; while LE135 and BMS-189453 are antagonists, and BMS-493 is an inverse agonist for RARβ. $K_D$ values are in parentheses. The heterodimer protein was incubated with each ligand at 8× concentrations for 1 h before assays started. The data points are plotted as mean ± SD from three technical replicates respectively

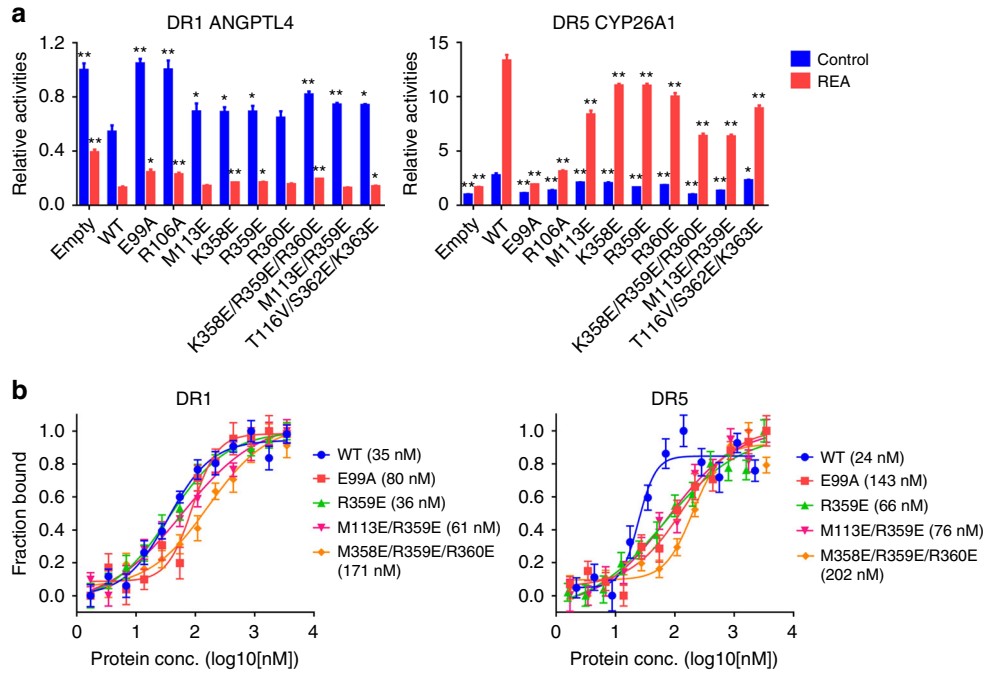

**Fig. 3** Effects of mutations at RARβ DBD–LBD interface on the transcription and DNA binding of RARβ–RXRα heterodimer. **a** Functional reporter assays testing the transcriptional activities of RARβ–RXRα. The wide-type (WT) RARβ and its mutants (majority at the interface) were tested for their transactivation on DR1 *ANGPTL4* (left) and DR5 *CYP26A1* (right) reporters, in the absence or presence of 1 μM REA. Each sample represents the average reading of cells from three wells, and data are shown as mean ± SD. The statistical differences between mutants (or the empty vector) and corresponding WT were calculated using the two-tailed t-test (\*P < 0.05; \*\*P < 0.01). **b** DNA-binding affinities of RARβ–RXRα heterodimer (WT and RARβ mutants) to DR1 DNA (left) and DR5 DNA (right) measured in the presence of REA and 9CR at 3× concentrations. $K_D$ values are in parentheses. The data points are plotted as mean ± SD from three technical replicates respectively

presence of RAR and RXR ligands REA and 9CR (Fig. 3b). Moreover, we used a 27-mer DR5 DNA, which had the same flanking bases as the 23-mer DR1. By using similar flanking sequences outside their direct-repeat elements, we could more clearly relate the binding affinities of DR1 vs. DR5 elements. The results suggest a $K_D$ of 35 nM for DR1 binding and 24 nM for DR5 binding, when WT RARβ and RXRα proteins were used together. As a reference point, we observed that single point mutation located precisely at the RARβ's DNA-binding helix (E99A) known to contact the DNA major groove within a half-site reduced the binding affinities for both DR1 (to 80 nM) and DR5 (to 143 nM). When we measured the effects of mutations located at the DBD–LBD interface (double and triple mutations), we also observed losses in the heterodimer's DNA affinities (three to tenfold) (Fig. 3b), suggesting that the integrity of RARβ DBD–LBD interface is important for maintaining the high-affinity binding mode of the heterodimer for both DR1 and DR5 elements.

**Structural differences among multi-domain NR complexes.** The RARβ–RXRα heterodimer described here represents the fourth distinct multi-domain NR complex studied to date using X-ray crystallography. To better understand the common features within the quaternary architectures of all these NR complexes, we directly compared the crystal structures of RARβ–RXRα, PPARγ–RXRα, HNF-4α homodimer, all of which were on DR1 DNA, and that of LXRβ–RXRα on DR4 DNA (Fig. 4a–d). With regard to RXR's subunit partners, we find that a single patch positioned on the LBDs of RARβ, PPARγ, and LXRβ (consisting of their α9 and α10 helices) appears to participate in forming domain–domain interfaces between LBDs and DBDs in these complexes (Fig. 4e–g). A similarly positioned patch within the LBD is also in the HNF-4α homodimer structure for forming LBD–DBD interactions (Fig. 4h).

Strikingly, as a group, these four crystallographically analyzed complexes do not conform closely to any of the rules for a common quaternary architecture advocated previously[7, 9]. Importantly, clear variations are observed in the relative locations of DBDs and LBDs within each complex (Fig. 4a–d and Supplementary Fig. 5), giving rise to a variety of distinct quaternary structures rather than a common or shared overall quaternary architecture. We also find that a previously suggested model for the quaternary structure of RARα–RXRα on DR1 DNA, based on the interpretation of solution studies by Rochel et al.[7] is inconsistent with the crystallographic, H/D-ex MS, and mutational studies presented here (Fig. 5). It is important to point out that those previous solution studies did not include enough experimental parameters to generate a reliable or detailed three-dimensional structural model with the reliability of crystallographic structures or H/D exchange studies that can render more precise information about local domain–domain structures and overall structural dynamics.

## Discussion

Our structural analysis of multi-domain RARβ–RXRα heterodimer has revealed an interdependent molecular architecture that involves LBD–LBD and DBD–DBD interfaces between RAR and RXR subunits, and an additional intramolecular DBD–LBD interface within the RARβ polypeptide. Both subunits were observed bound to their unique ligands, and were separately bound to peptide segments of a p160 coactivator protein. The observed polarity of the complex on DR1 is established with RAR binding on the upstream AGGTCA DNA half-site and RXR binding on the downstream half-site. The overall quaternary structure that was visualized shows that each subunit maintains access for binding to an LXXLL motif and a half-site within the DNA response element, without posing any physical restrictions

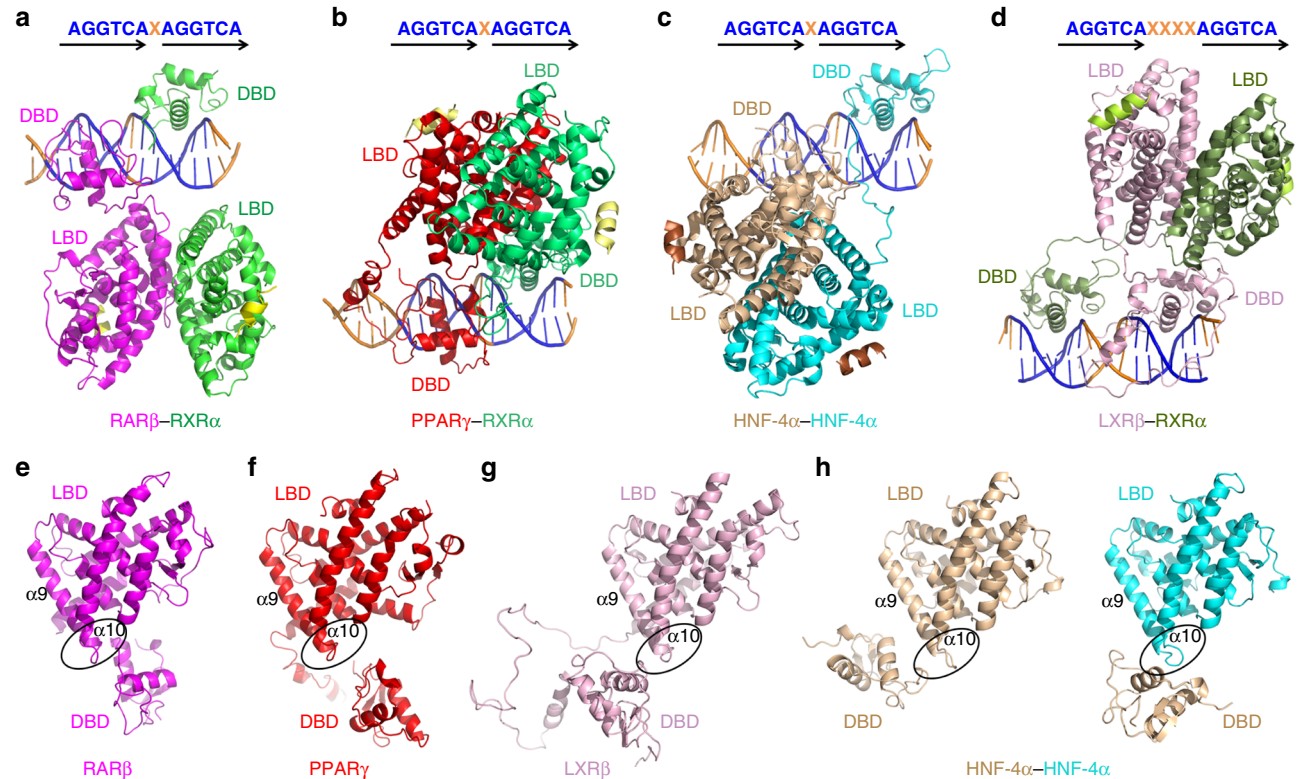

**Fig. 4** The different architectures of multi-domain NRs on DNA. **a–d** Crystal structures of RARβ–RXRα heterodimer (**a**), PPARγ–RXRα heterodimer (**b**), and HNF-4α homodimer (**c**) on DR1 DNA, and that of LXRβ-RXRα heterodimer on DR4 DNA (**d**). The AGGTCA sites are colored in blue for each structure. **e–h** Relative positions of LBDs and DBDs of RARβ (**e**), PPARγ (**f**), LXRβ (**g**), and HNF-4α (**h**). All the LBDs are in the same orientation, with their α9 and α10 helices labeled, and their "patch" regions highlighted with clear ovals. For HNF-4α homodimer, one DBD is shown near to the LBDs from the same monomer (in wheat color) or from the other one (in cyan)

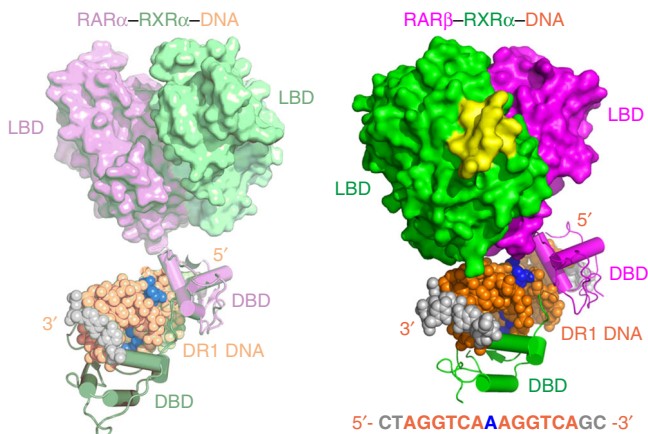

**Fig. 5** Significant differences between a previously proposed structural model of RARα–RXRα heterodimer on DR1 by Rochel et al.[7, 35] (left) and the current crystal structure of RARβ–RXRα on DR1 (right). The proteins and DNA are colored in the same way as in both figures. The DBD domains from two complexes are positioned in the same orientation to clearly highlight the overall differences in quaternary structures. Note that in these two models, the LBD–LBD segments have clearly different (rotational and translational) positionings with respect to both their DR1 DNA and DBDs

or barriers on the other subunit's access or interactions. While most previous structural studies showing the interactions of the DBD with DNA or the interactions of two subunits' LBDs are clearly validated and reaffirmed in the structure of their multi-domain complex presented here[13], the correct locations of all the

domains within the quaternary structure, the RARβ protein's DBD-LBD contacts, and the physical path for allosteric transmission through the RAR protein were not previously known or accurately described.

The current structural findings, when examined together with previously reported structures for multi-domain NR complexes, point to several important concepts about the nature and extent of domain—domain interactions within NR complexes. RXR is found to display flexibility within its own polypeptide, and not favoring a fixed physical connection involving its own DBD and LBD domains (Supplementary Fig. 5). This polypeptide flexibility naturally facilitates an adaptive function in forming heterodimers with a variety of other NRs. The flexibility is also a key to the facile adaptation of RXR heterodimers to a variety of distinctly spaced DNA direct-repeats. Interestingly, the intramolecular DBD–LBD interactions within the RARβ subunit appear to be required for RARβ–RXRα functions through both DR1 and DR5 response elements, suggesting that RARβ's internal domain—domain interactions are predisposed based on a favorable interaction of its LBD patch and DBD surface.

Our studies also find that for RXR's distinct subunit partners, a single patch positioned on their LBDs consisting of their α9 and α10 helices acts as a common surface for forming domain–domain interactions with DBDs in these complexes (Fig. 4e–h). The amino acids within this patch are variable and not conserved, yet the physical location of the patch is conserved within all the RXR partners seen within these crystal structures. Finally, the H/D-ex MS and DNA-binding studies suggest that the DBD–LBD physical couplings involving these patches can allow for effective allosterical signal transmission from the LBD to the DBD. The LBD–DBD junction allows information about the type of ligand bound within

the LBD to be allosterically sent to the DBD and shape the overall DNA affinity of the NR complex. This finding extends our previous characterization of functional coupling between LBDs and DBDs observed in the context of the HNF-4α homodimer where MODY1 mutations located within LBD were seen to project an allosterically transmitted influence on the function of the DBDs and the DNA-binding affinity of that complex[5].

Understanding the precise molecular underpinnings of NR function requires the direct high-resolution visualization of multi-domain and functionally dimerized NRs bound to DNA and also with the constellation of transcriptional coregulators that interact upon a specific signal or stimulus. The ability to directly visualize these types of complexes for crystallographic studies has been largely hampered by abilities to generate the required full-length proteins in pure and stable forms, and the further difficulties in forming diffracting crystals when each of these proteins has large disordered regions within their polypeptides. Due to these limitations, the large majority of structural studies conducted over the past three decades for the NR family involved only isolated single domains; therefore, failing to reveal the complexities and variations of their quaternary architectures and the physical pathways for functional communications between domains. The current structure and the three previous co-crystal structures of NRs that used multi-domain proteins and dimeric complexes on DNA show that the overall quaternary states in this family are quite diverse and do not conform to a previous set of rules suggested from an interpretation of solution-based studies[7]. Moreover, the key interactions between LBDs and DBDs that are consistently seen crystallographically in four structures to date, were missing and incorrectly interpreted in a set of previously suggested models[7] (Fig. 5).

Moving forward, significant gaps still remain in our detailed understandings of how pairs of protein coactivators and other transcriptional components in their full-length forms assemble on these receptors. Given the limitations in crystallographically observing more elaborate and larger scale multi-protein NR structures, cryo-EM studies may prove to be increasingly useful. Cryo-EM studies recently conducted on the estrogen receptor[11] has extended our understanding of coregulator binding in the context of more complete and accessorized NR complexes. That study also showed the mode of further involvement of ligand-independent transactivation function 1 (AF-1) domains located at the N-termini of these receptors, which have not been success-fully visualized in any of the NR crystal structures to date. Future co-crystal and cryo-EM studies focused on more complete protein–protein and protein–DNA complexes will be needed to fully understand the wider spectrum of molecular components and interactions underlying NR functions.

## Methods

**Protein expression and purification.** Human RARβ (GenBank accession: AAH60794.1, residues 73–414) and RXRα (GenBank accession: BAH02296.1, residues 98–462) were cloned into the vectors pET28a (N-His tagged) and pET20b (No tag), respectively. PCR primer sequences used in this study are summarized in Supplementary Table 1. Then the two recombinant plasmids were co-transformed into BL21-CodonPlus (DE3)-RIL competent cells (Agilent Technologies). After overnight culture at 16 °C with 0.5 mM IPTG, the cells were harvested and lysed by sonication in buffer containing 20 mM Tris (pH 7.5), 500 mM NaCl, 20 mM imidazole, 10 μM 9CR, and 10% (v/v) glycerol. After centrifugation, cell lysate supernatant was loaded on a Ni-column packed with His-Bind resin (Novagen), which was then washed with five bed volumes of lysis buffer and eluted with buffer containing 20 mM Tris (pH 7.5), 500 mM NaCl, 300 mM imidazole, 5% (v/v) glycerol. Eluted protein was diluted with 20 mM Tris (pH 7.5) to about 100 mM NaCl concentration, and loaded on a S-column packed with SP Sepharose (GE Healthcare). After washing with three bed volumes of the same diluted buffer, proteins were eluted with buffer containing 20 mM Tris (pH 7.5), 400 mM NaCl, 5% (v/v) glycerol. The final purification step was conducted on the Superdex 200 pg gel-filtration column (GE Healthcare) with running buffer containing 20 mM Tris (pH 7.5), 150 mM NaCl, 5% (v/v) glycerol. Purified RARβ–RXRα protein complex

was combined with the synthetic double-strand DR1 DNA (forward: 5′-CTAGGTCAAAGGTCAGC-3′ and reverse: 5′-GCTGACCTTTGACCTAG-3′) in a 1:1.5 ratio. After a further gel-filtration step in buffer containing 20 mM Tris (pH 7.5), 150 mM NaCl and 5% (v/v) glycerol, the protein–DNA complex fractions were supplemented with 10 mM DTT, as well as 3× molar ratio of REA (all-trans retinoic acid), 9CR (9-cis retinoic acid) and the SRC-2 LXXLL peptide (EKHKILHRLLQDSY).

**Crystallization and X-ray data collection.** Crystallization of the RARβ–RXRα–DNA complex was carried out using the hanging drop vapor dif-fusion method at 4 °C, by mixing equal volume of protein–DNA complex (4 mg ml$^{-1}$) and reservoir solution containing 0.1 M MES (pH 6.5), 25% (v/v) PEG300. Before flash frozen in liquid nitrogen, crystals were soaked in reservoir plus 30% (v/v) glycerol as the cryoprotectant. Diffraction data were collected at the Argonne National Laboratory SBC-CAT 19ID beamline at 100 K, and processed using the HKL3000 program[23].

**Structure determination and refinement.** The structure of RARβ–RXRα–DNA complex was solved by molecular replacement with the program Phaser[24], using their DBD and LBD heterodimer structures (PDB: 1DSZ[17] and 1XDK[18]) as the search models. Further manual model building was facilitated using Coot[25], combined with the structure refinement using phenix.refine[26]. The diffraction data and final statistics are summarized in Table 1. The Ramachandran statistics, as calculated by MolProbity[27], are 95%/0% (favored/outliers). All the structural fig-ures were prepared using PyMOL (http://www.pymol.org).

**Fluorescence polarization DNA-binding assay.** The fluoresceinated double-strand DNA was prepared by annealing 6-FAM 5′-labeled forward strands (5′-AAACTAGGTCAAAGGTCAGAAAG-3′ for DR1, and 5′-AAACTAGGTCACC-GAAAGGTCAGAAAG-3′ for DR5) with the unlabeled reverse strands (5′-CTTTCTGACCTTTGACCTAGTTT-3′ for DR1, and 5′-CTTTCTGACCTTTCGGTGACCTAGTTT-3′ for DR5) in the buffer consisting of 10 mM Tris (pH 7.5), 1 mM EDTA and 2 mM MgCl₂. 2 nM DNA was incubated with purified proteins for 30 min for the binding assay. Final protein concentra-tions were varied by serial dilution in binding buffer (20 mM Tris pH 7.5, 200 mM NaCl and 10 mM DTT). The fluorescence polarization signals were recorded using black 96-well plates on FlexStation 3 (Molecular Devices)and converted to "fraction bound", before fitting the curves sigmoidally in GraphPad Prism 7 to get the $K_D$ values[28].

**Luciferase reporter assay.** Full-length human RARβ was cloned into the pCMV-Tag4 vector, and the subsequent site-directed mutagenesis was confirmed by DNA sequencing. For the experiments using ANGPTL4 (DR1) reporter, HEK293T cells (ATCC CRL-3216, not authenticated nor tested for mycoplasma contamination) were seeded in 24-well plates, and 1 day later transfected with 100 ng of human RXRα, 100 ng of luciferase reporter, 1 ng of pRL (control Renilla luciferase) and 100 ng of pCMV-Tag4-RARβ (WT, mutants or empty plasmid) by using 0.6 μL jetPRIME regent (Polyplus-transfection SA) for each well. For the experiments using CYP26A1 (DR5) reporter, HEK293T cells were prepared similarly and transfected with 100 ng of luciferase reporter, 1 ng of pRL (control Renilla luci-ferase) and 200 ng of pCMV-Tag4-RARβ (WT, mutants or empty plasmid) by the same regent. After 6 h transfection, medium was refreshed with 1 μM REA; and another 24 h later, luciferase activity was measured using the Dual-Glo Luciferase Assay System (Promega). Final data were normalized by the relative ratio of firefly and Renilla luciferase activity. All experiments were repeated independently for at least twice.

**Hydrogen/deuterium exchange mass spectrometry experiments.** Hydrogen–deuterium exchange experiments were performed using our in-house deuterium exchange system[29, 30], in which enzymatic digestion, peptide separation and MS analysis was fully automated. To initiate the HDX reaction, 6 μl of pre-chilled protein stock solution at 5.4 μM (free RARβ–RXRα protein complex, RARβ–RXRα protein complex combined with DNA and agonist or antagonist) was diluted into 18 μl D₂O buffer (8.3 mM Tris, 150 mM NaCl, in D₂O, pD$_{READ}$ 7.2), and incubated at 0 °C for 30, 100, 300, 1000, 3000, and 10,000 s. At indicated times, 6 μl of ice cold acidic buffer (0.8 M GuHCl, 600 mM NaH₂PO₄, 50% (v/v) Glycerol, pH 2.4) was added to quench the exchange reaction. Quenched samples were frozen on dry ice and passed over an in-house made immobilized pepsin column (16 μl bed volume) on ice in H₂O contain 0.05% (v/v) trifluoroacetic acid at a flow rate of 20 μl min$^{-1}$ for pepsin digestion. The proteolytic products were collected on a C18 trap column for desalting (Optimize Technologies, Magic C18 AQ, 0.2 × 2 mm) and eluted for separation on a Michrom Magic C18 (3 μm, 0.2 × 50 mm, 200 Å) with a 30 min linear acetonitrile gradient (6.4–38.4%). The effluent was subject to an OrbiTrap Elite mass spectrometer (Thermo Fisher Scientific) for MS analysis. The instrument settings were optimized for HDX analysis as previously reported[31]. Peptide identification was performed using Proteome Discoverer (Thermo Fisher Scientific) and deuterium level of each peptide was determined by using HDEXaminer (Sierra Analytics Inc., Modesto, CA). In addition, HDX analysis was also carried out on non-deuterated and fully deuterated samples to correct back-

exchange[32]. A peptide coverage map for the RARβ protein used in the HDXMS studies and individual uptake curves (including the regions highlighted in Fig. 2b, c) are shown in Supplementary Fig. 6.

**Sequence analysis**. The protein sequence alignments were generated on the Clustal Omega sever[33], and were subsequently processed by the ESPrint 3.0 program[34] for figures.

**Data availability**. Coordinates and structure factors have been deposited in the Protein Data Bank under accession code PDB 5UAN. Other data are available from the corresponding author upon reasonable request.

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

## Acknowledgements

We thank Z. Simandi and L. Nagy for the human RXRα expression and luciferase reporter plasmids, members of the Structural Biology Center at Argonne National Laboratory for their help with data collection at the 19-ID beamline. This work was supported by the National Institutes of Health grants (R01GM117013 and R01GM120532) to F.R., NIH grants (1U19AI117905, R01GM020501, R01AI101436) to S.L., and the Qilu Young Scholar funding from Shandong University (11200086963072) to D.W.

## Author contributions

V.C. purified the proteins, carried out crystallization, and performed the biochemical assays. D.W. solved the structure, conducted cell-based experiments, and wrote the manuscript. S.L. performed the hydrogen/deuterium exchange mass spectrometry experiments. N.P. produced the expression and mutation constructs. Y.K. collected and processed synchrotron diffraction data. F.R. supervised the work and wrote the manuscript.

## Additional information

**Competing interests:** The authors declare no competing financial interests.

