## [Peer Review file · Nature Communications]

Reviewers' comments:

Reviewer #1 (Remarks to the Author):

The authors present here the first crystal structure of the multi-domain RAR β -RXR α heterodimer bound to DNA, in presence of agonist ligands and coactivator peptides. From this structure and from data derived from H/D-ex MS experiments and functional transcriptional studies, they describe the relative organization of RAR and RXR functional domains and the interfaces between the domains.

Nevertheless, the conclusions derived from this structural work are small incremental advances as this new structure does not give strong new molecular details to decipher the mechanism of action of RAR-RXR heterodimer.

In particular, in the results section, authors often provide more a discussion than a real interpretation of results:

- The paragraph from lines 103 to 114 looks like a discussion and the results presented in this manuscript do not allow to choose between the two models (one coactivator or two coactivator molecules bound to the heterodimer). In addition, the presence of two coactivator peptides has already been shown in the structure of RAR β -RXR α LBDs in complex with coactivator peptides (Ref 13).
- Another example is from lines 163 to 166 where the authors discuss about the presence of the residue S362, a target of phosphorylation, at the interface between LBD and DBD, but do not give any new results about the consequences of this particular location.
- Another very clear example spreads from lines 179 to 183 that is really a discussion and not a rigorous interpretation of the data.

Secondly, the conclusions are not strongly supported by experimental data:

- In lines 140-141, the authors wrote that "therefore, we find that the physical connection between the LBD and DBD of RAR β produces a path of signal transmission between these domains." However, to our mind, the "signal transmission" is not demonstrated by this single HDX experiment. To support this conclusion, the authors should (1) test their hypothesis with more ligands, (2) show that there is a functional relationship between the binding of one type of ligand and the affinity of the heterodimer for the DNA and (3) that disrupting this communication path (the BDB-LBD interface) by specific mutations would have a functional effect. Similarly, the results could suggest "a physical path for allosteric signal transmission through the RAR protein" (Lines 195-196) but additional data are necessary to validate this hypothesis.
- It is not clear which experimental results give "strong indication on the role of the DNA response elements in facilitating and stabilizing the DBD-LBD interactions" (lines 206-207). The functional transcriptional studies and mutagenesis are not convincing enough. From Figure 3 and Lines 153-154: why REA is active on DR1, even with the empty vector? Why REA is still active with the mutants E99A and R106A? (The activation is lost on DR5). The effects of the mutants (even the triple mutants) are modest and overall the results shown in this figure do not fully convince this reviewer of the functional relevance of the observed interface. In Figure 3, the authors should add statistic values to really confirm significant differences?
In addition, it would have been of high interest to measure the DNA binding affinity of the different mutants to fully validate that "the intramolecular DBD-LBD interaction within the RAR β subunit appear to be required for RAR-RXR functions through both DR1 and DR5 response elements" (lines

203-204).

Additional comments:

- Table 1 and quality of the structural data. In the highest resolution shell, data are of low completeness.

The structure is of medium resolution and the quality of the map is difficult to judge from the one presented in Supplementary Figure 1. The authors should also show the Fo-Fc and 2Fo-Fc maps of both ligands.

- Line 72: errors on KD values are surprisingly very low, especially when looking at the error bars of the corresponding curves (Fig 1b).

- Line 73-74: the affinity of RAR β -RXR α heterodimer for DR1 is surprisingly higher than for DR5. It is established that RXR-RAR heterodimers have a stronger affinity for DR5. How the authors justify this observation? In this experiment, it would have been better to use DR1 and DR5 DNAs with the same number of flanking bases to allow a true comparison of the affinities for DR1 and DR5. In addition, the heterodimer was purified in absence of ligand. It would have been informative to test the influence of the ligands REA and 6CR on DNA binding.

- Line 75: Was the full-length RAR β -RXR α heterodimer used for crystallization?

- Lines 86-87: The authors should add a supplementary figure to show the details the interfaces between RAR and RXR.

- Lines 88-89: The authors should give a quantitative information like a rmsd value of this similarity.

- Lines 119-120: The interaction LBD-DBD in RAR β appears rather non-specific. This contradicts the "sequence conservation" in RAR isotypes.

Reviewer #2 (Remarks to the Author):

Dear Authors,

The work presented here on how different domains of RAR and RXR proteins interact using crystallography, HDXMS and functional assays is well thought out and executed and brings novel information regarding the dynamics and allostery in these proteins.

I request a few additional details on the HDXMS studies to be added to the supplementary material. was specifically asked to review to the HDXMS part of this paper.

The methods section pg 13, line 20 address volume of protein and complex, it would be useful to also present the concentrations of the protein and complexes used in the study.

Could the authors comment on the use of a 1:3 dilution for the HDXMS studies and either give detailed explanation of the back exchange experiment and correction or a reference to point to the same.

Please add a peptide coverage map for the HDXMS studies and individual uptake curves, especially for the regions that have been highlighted in fig 2b and 2c.

Thank you

RAR β -RXR α heterodimer has revealed an interdependent molecular architecture that involves LBD-LBD and DBD-DBD interfaces between RAR and RXR subunits, and an additional intramolecular DBD-LBD interface within the RAR β protein itself to assess their level of overall variations and to pinpoint specific sites where domain-domain junctions are formed in each case.

Reviewer #3 (Remarks to the Author):

The manuscript by Chandra et al. describes a valuable new multi-domain structure of a previously unsolved complex, that of the RAR β -RXR α heterodimeric nuclear receptor (NR). Given that ligands exist that are able to modulate the gene expression programs of these transcription factors, which are involved in several diseases, it is of the utmost importance to understand the structure-function relationships underlying their pharmacological outcomes and how these proteins exert their cellular functions under physiological or pathological conditions. There are only very limited number of crystal structures of multi-domain NRs solved using X-ray crystallography (the heterodimer of PPAR γ +RXR α , PDB codes 3E00, 3DZU and 3DZY, all of them including different ligands; the homodimer of HNF4 α in PDB code 4IQR; the heterodimer of LXR β + RXR α in PDB code 4NQA). Comparative results of these previous crystal structures with that of the structures solved using low-medium resolution biophysical techniques such as SAXS, have revealed that there is a diversity of NR quaternary structures rather than a unified common model of domain spatial arrangements. Two of the above-mentioned structures, which were solved by the same group (Chandra et al. Nature 2008; et al. Chandra et al. Nature 2013), represented a major breakthrough in the understanding of NR structure and function, besides showing for the first time that communication existed between NR functional modules that have not been previously envisioned from the isolated domain structures. This finding opened possibilities to the understanding of both close-contact or long-range (allosteric) inter-domain signal transmission. The manuscript now presents further evidence of the diversity of quaternary structures of a multi-domain NR. Besides, the manuscript is easy to follow and clearly written both for general readers as well as experts familiar with the architectures of multi-domain NRs.

Overall, I strongly support the publication of this manuscript in Nature Communications, the manuscript is of high quality and the structure is relevant to both structural and Biomedical applications.

I have however, some suggestions for improving the text and understanding of their findings, as indicated below:

-On page 2 lines 33-35. The author lists the multi-domain NRs solved by X-ray crystallography, I would like them to add the PDB codes of each one of them to help out the reader to find the information needed to find them right away.

-I suggest to add between lines 33 and 36 a list of other multi-domain NRs solved using other biophysical techniques (SAXS, SANS) so that it is also easier to understand where the comments in the Results section – lines 93-1114 refer to. There are two references that should be also added in this manuscript, Rastinejad et al. TiBS (2015) and Moras D et al. in TiBS (2015).

-On page 3 line 60, it appears mentioned for the first time the technique hydrogen-deuterium exchange mass spectrometry, which the authors mention later on as either H/D-ex MS, H/D-ex or

HDX. I suggest they use only one of these ways during the entire manuscript.

-On page 3 lines 60-61 the authors refer to the fact that HDX can aid in determining whether there is allosteric signal transmission across the heterodimer to distal domains. I would like the authors to guide here much more the reader as to what do they mean by "distal domains", to the ones belonging to one of the polypeptides or communication across the polypeptide chains.

-On page 4 line 71 please refer to the technique here used to measure the binding affinities to the DNA half-sites.

-On page 4 line 78, what NR box of SRC-2 have they used for the complex formation?

-On page 4 line 78, the search model used for molecular replacement could be already mentioned here.

- On page 4 line 84, I would suggest a more descriptive way to explain how the two domains of RXRa are spatially displaced to each other.

-On pages 4 and 5 lines 84-85 and also in Figure 1c. Include the amino acid numbers that belong to the hinge domains that either have very poor or non-existent electron densities connecting the RXRa domains. This information can be added in the figure 1 legend so that the readers will understand what the dashed lines mean in the figure.

- On page 5 line 94 the authors refer to helix 12, it will be appreciated if this helix is labeled in some of the figures.

- Figure 1a, additional remarks need to be added to the schematic cartoon depicting RAR β , the authors mention residue 414 as the end of the construct that has crystallized. They should add the number 448 at the end of this cartoon to show that this polypeptide misses C-terminal residues.

-Figure 1d, the DBD-LBD interface participating residues are shown as stick models. Add more level of detail to indicate the kind of contacts, hydrogen-bonds and the buried surface of each domain.

-For non-experts on the H/D-ex MS patterns observed, I would suggest on page 6 lines 127-129 to clarify what are we expected to see on figure 2.

-On page 6 line 127, the authors state that "clear changes within the H/D-ex MS patterns" are observed. Please clarify what kind of changes, the details of what they refer to.

-On page 7 line 152 onwards, a series of mutations at the DBD-LBD junction have been studied. What was the rationale behind the chosen point-mutations, were they only a biochemical exercise or also some of these mutations have been associated to pathology?

-On page 8 line 174, the authors state that significant variations have been clearly observed in the relative locations of the DBDs and the LBDs. Have the authors calculated r.m.s.d? Please give more detail here for structural biologists.

-On page 9 line 178, the structures solved using SAXS are different than the ones solved using X-ray crystallography. The authors use the term "inconsistent". Would it be possible that the protein in solution and the protein conformation that crystallized present major differences, which can be attributed solely to the fact that different conformational minima or preferred conformations only occur when the protein is in solution, and when in the form of a crystal the neighboring molecules (crystal packing) contribute to these major differences. Would it be possible that this NR conformational variability makes both models possible according to the methods used and be both

likely to exist in different cellular conditions? Does the model obtained by one biophysical technique necessarily invalidate the model obtained using the other technique, even if we leave aside the resolution obtained by the most powerful technique?

-On page 13 line 283. How many independent experiments and measurements were done, please mention it here.

Response to reviewers:

We thank all three reviewers for their careful evaluation, encouragement, and constructive guidance in improving our manuscript. We have carefully addressed all their suggestions, including carrying out several new experiments.

Reviewer #1 (Remarks to the Author):

The authors present here the first crystal structure of the multi-domain RAR β -RXR α heterodimer bound to DNA, in presence of agonist ligands and coactivator peptides. From this structure and from data derived from H/D-ex MS experiments and functional transcriptional studies, they describe the relative organization of RAR and RXR functional domains and the interfaces between the domains.

Nevertheless, the conclusions derived from this structural work are small incremental advances as this new structure does not give strong new molecular details to decipher the mechanism of action of RAR-RXR heterodimer.

In particular, in the results section, authors often provide more a discussion than a real interpretation of results:

- The paragraph from lines 103 to 114 looks like a discussion and the results presented in this manuscript do not allow to choose between the two models (one coactivator or two coactivator molecules bound to the heterodimer). In addition, the presence of two coactivator peptides has already been shown in the structure of RAR β -RXR α LBDs in complex with coactivator peptides (Ref 13).

The reviewer has a good point, and we have improved the clarity of this section to be more consistent with our data and its interpretation. See page 5 bottom where we discuss the binding of two LXXLL peptides to an RAR-RXR heterodimer versus binding of one peptide only. The original text mistakenly referred to this issue as a question of two versus one *coactivators* binding to RAR-RXR (but should have referred instead to one versus two LXXLL peptides). The data and citations we include are very clearly able to distinguish between one or two LXXLL peptides bound (crystallographic observations) and refutes the alternative proposal that had been suggested in another paper saying that one LXXLL peptide binding to one subunit would disfavor the binding of the second peptide to the second subunit.

- Another example is from lines 163 to 166 where the authors discuss about the presence of the residue S362, a target of phosphorylation, at the interface between LBD and DBD, but do not give any new results about the consequences of this particular location.

We deleted this section from the results section.

- Another very clear example spreads from lines 179 to 183 that is really a discussion and not a rigorous interpretation of the data.

We deleted this section also, to focus the writing more on our own data/results.

Secondly, the conclusions are not strongly supported by experimental data:

- In lines 140-141, the authors wrote that "therefore, we find that the physical connection between the LBD and DBD of RAR β produces a path of signal transmission between these domains." However, to our mind, the "signal transmission" is not demonstrated by this single HDX experiment. To support this conclusion, the authors should (1) test their hypothesis with more ligands, (2) show that there is a functional relationship between the binding of one type of ligand and the affinity of the heterodimer for the DNA and (3) that disrupting this communication path (the DBD-LBD interface) by specific mutations would have a functional effect. Similarly, the results could suggest "a physical path for allosteric signal transmission through the RAR protein" (Lines 195-196) but additional data are necessary to validate this hypothesis.

We have carried out new experiments as the reviewer requested, testing the DNA binding affinity of RAR β -RXR α heterodimer in the presences of multiple RAR ligands (new Fig. 2d), and these data showed that the DNA binding affinity of RAR β -RXR α does vary in the presence of different ligands, and in general these affinities are weaker with agonists than those with antagonist. Description of the data was added in the text on Page 7 Lines 18-23.

In addition, we also carried out new experiments requested, testing the functional effects of mutations positioned at the RAR β DBD-LBD interface on DNA binding affinities of RAR β -RXR α for both DR1 and DR5 elements (new Fig. 3b), as described on Page 9 Lines 3-15.

- It is not clear which experimental results give "strong indication on the role of the DNA response elements in facilitating and stabilizing the DBD-LBD interactions" (lines 206-207).

The statement about "strong indication on the role of the DNA response elements in facilitating and stabilizing the DBD-LBD interactions" has been removed.

The functional transcriptional studies and mutagenesis are not convincing enough. From Figure 3 and Lines 153-154: why REA is active on DR1, even with the empty vector? Why REA is still active with the mutants E99A and R106A? (The activation is lost on DR5). The effects of the mutants (even the triple mutants) are modest and overall the results shown in this figure do not fully convince this reviewer of the functional relevance of the observed interface. In Figure 3, the authors should add statistic values to really confirm significant differences?

The effects that REA is active on DR1 with the empty vector, as well as with mutants E99A and R106A are due to the background activities in the HEK293T cells, which came from intrinsic RARs and/or RXRs activated by REA, and were more active on DR1 reporter than on DR5. To make the functional changes caused by mutations clearer in Fig. 3a, we have added statistics to the data of mutants in comparison to the wild-type. The fact that RAR-RXR repress transcription from DR1 elements while enhancing transcription from DR5 elements is well known in the field. The citation below is relevant for this issue. We believe the new biochemical (DNA-binding studies) that we have included in the manuscript within the same panel (3b) using

both DR1 and DR5 elements in the context of the same point mutations, will provide additional support about the functional importance of the DBD-LBD domain-domain interaction sites.

Nature 1995: 377 (451-4), Kurokawa R., Soderstrom M., *et al.* Polarity-specific activities of retinoic acid receptors determined by a co-receptor:
<https://www.ncbi.nlm.nih.gov/pubmed/7566126>

In addition, it would have been of high interest to measure the DNA binding affinity of the different mutants to fully validate that “the intramolecular DBD-LBD interaction within the RAR β subunit appear to be required for RAR-RXR functions through both DR1 and DR5 response elements” (lines 203-204).

As we mentioned just above, we did carry out new experiments to measure the DNA binding affinities of different RAR β mutants in complex with RXR α for both DR1 and DR5. As suggested by the referee, we also took steps to use DNA duplexes with the same flanking bases in both DR5 and DR1 (new Fig. 3b). The results point to the importance of DBD-LBD interface of RAR β for binding of both DNA elements, since mutations introduced at the interfaces of RAR β 's LBD and DBD did weaken the DNA binding of the heterodimer for both DR1 and DR5 elements (Page 9 Lines 3-15). As the reviewer will note, we made single, double and triple point mutations positioned near the observed DBD-LBD interface of RAR β -RXR α .

Additional comments:

- Table 1 and quality of the structural data. In the highest resolution shell, data are of low completeness.

The structure is of medium resolution and the quality of the map is difficult to judge from the one presented in Supplementary Figure 1. The authors should also show the Fo-Fc and 2Fo-Fc maps of both ligands.

The Fo-Fc and 2Fo-Fc maps of both ligands are now included in new Supplementary Fig.1.

- Line 72: errors on KD values are surprisingly very low, especially when looking at the error bars of the corresponding curves (Fig 1b).

The old Fig. 1b has been removed due to redundancy to the other binding data shown later in the manuscript. As we discussed above, we carried out new studies in Fig. 3b where we used the same flanking sequences on both DR1 and DR5 (see WT affinities). These experiments were carried out with appropriate numbers of replicates as shown.

- Line 73-74: the affinity of RAR β -RXR α heterodimer for DR1 is surprisingly higher than for DR5. It is established that RXR-RAR heterodimers have a stronger affinity for DR5. How the authors justify this observation? In this experiment, it would have been better to use DR1 and DR5 DNAs with the same number of flanking bases to allow a true comparison of the affinities for DR1 and DR5

The issue of standardizing the flanking residues on DR1 and DR5 was addressed as discussed above. The DR1 and DR5, when carrying the same flanking residues, have similar affinities for RAR β -RXR α (with slightly weaker affinity of 35 nM K_D for DR1 than the 24 nM K_D for DR5).

In addition, the heterodimer was purified in absence of ligand. It would have been informative to test the influence of the ligands REA and 6CR on DNA binding.

The effects of different agonists (including REA and 9CR) and antagonists on DNA binding of the RAR β -RXR α heterodimer is now shown in our new Fig. 2d. And in the experiments measuring the binding affinities of RAR β mutants to DR1 and DR5 DNAs, all the protein complexes were tested in the presence of REA and 9CR.

- Line 75: Was the full-length RAR β -RXR α heterodimer used for crystallization?

No, we used both RAR β and RXR α multi-domain constructs including the DBD, hinge and LBD regions. We clearly state this in the beginning of the results section. Further details about the starting and ending amino-acid positions in our crystallization constructs is also shown in Fig. 1a.

- Lines 86-87: The authors should add a supplementary figure to show the details the interfaces between RAR and RXR.

The details of the molecular interactions between RAR β and RXR α (at both LBD and DBD dimer interfaces) are now shown within Supplementary Fig. 2a.

- Lines 88-89: The authors should give a quantitative information like a rmsd value of this similarity.

This information has been added in the text on Page 5 Lines 15-16.

- Lines 119-120: The interaction LBD-DBD in RAR β appears rather non-specific. This contradicts the "sequence conservation" in RAR isotypes.

This DBD-LBD interface in RAR β is mediated by hydrogen bonds and hydrophobic interactions between residues on both domains and these are specific. As we show in Supplementary Fig. 4a), the sequence conservation in RAR isotypes at these junctional nodes is quite high too.

Reviewer #2 (Remarks to the Author):

Dear Authors,

The work presented here on how different domains of RAR and RXR proteins interact using crystallography, HDXMS and functional assays is well thought out and executed and brings novel information regarding the dynamics and allostery in these proteins.

RAR β -RXR α heterodimer has revealed an interdependent molecular architecture that involves LBD-LBD and DBD-DBD interfaces between RAR and RXR subunits, and an additional intramolecular DBD-LBD

interface within the RAR β protein itself to assess their level of overall variations and to pinpoint specific sites where domain-domain junctions are formed in each case.

I request a few additional details on the HDXMS studies to be added to the supplementary material. was specifically asked to review to the HDXMS part of this paper.

The methods section pg 13, line 20 address volume of protein and complex, it would be useful to also present the concentrations of the protein and complexes used in the study.

The protein complex concentrations were 5.4 μ M (0.45 mg/ml). This information has been added in the methods section accordingly.

Could the authors comment on the use of a 1:3 dilution for the HDXMS studies and either give detailed explanation of the back exchange experiment and correction or a reference to point to the same.

The 1:3 ratio we used is just to balance the protein amount and maximum deuteration level so that we have enough amount protein to get best sequence coverage and enough maximum deuteration level for monitoring HDX profile changes of each peptide. We used equilibrium-deuterated samples as control to correct the back exchange. The following literature has been added as a new reference.

Li S, Tsalkova T, White MA, Mei FC, Liu T, Wang D, Woods VL Jr, Cheng X. (2011) Mechanism of intracellular cAMP sensor Epac2 activation: cAMP-induced conformational changes identified by amide hydrogen/deuterium exchange mass spectrometry (DXMS). *J Biol Chem.* 286:17889-97.

Please add a peptide coverage map for the HDXMS studies and individual uptake curves, especially for the regions that have been highlighted in fig 2b and 2c.

We added this information within Supplementary Fig. 6 showing the coverage map (a) and individual uptake curves (b) for the selected peptides in highlighted regions in Fig. 2b and 2c.

Reviewer #3 (Remarks to the Author):

The manuscript by Chandra et al. describes a valuable new multi-domain structure of a previously unsolved complex, that of the RARb-RXRa heterodimeric nuclear receptor (NR). Given that ligands exist that are able to modulate the gene expression programs of these transcription factors, which are involved in several diseases, it is of the utmost importance to understand the structure-function relationships underlying their pharmacological outcomes and how these proteins exert their cellular functions under physiological or pathological conditions. There are only very limited number of crystal structures of multi-domain NRs solved using X-ray crystallography (the heterodimer of PPARg+RXRa, PDB codes 3E00, 3DZU and 3DZY, all of them including different ligands; the homodimer of HNF4a in PDB code 4IQR; the heterodimer of LXRb + RXRa in PDB code 4NQA). Comparative results of these previous crystal structures with that of the structures solved using low-medium resolution biophysical techniques such as SAXS, have revealed that there is a diversity of NR quaternary structures rather than

a unified common model of domain spatial arrangements. Two of the above-mentioned structures, which were solved by the same group (Chandra et al. Nature 2008; et al. Chandra et al. Nature 2013), represented a major breakthrough in the understanding of NR structure and function, besides showing for the first time that communication existed between NR functional modules that have not been previously envisioned from the isolated domain structures. This finding opened possibilities to the understanding of both close-contact or long-range (allosteric) inter-domain signal transmission. The manuscript now presents further evidence of the diversity of quaternary structures of a multi-domain NR. Besides, the manuscript is easy to follow and clearly written both for general readers as well as experts familiar with the architectures of multi-domain NRs.

Overall, I strongly support the publication of this manuscript in Nature Communications, the manuscript is of high quality and the structure is relevant to both structural and Biomedical applications.

I have however, some suggestions for improving the text and understanding of their findings, as indicated below:

-On page 2 lines 33-35. The author lists the multi-domain NRs solved by X-ray crystallography, I would like them to add the PDB codes of each one of them to help out the reader to find the information needed to find them right away.

The PDB codes of these structures have been added accordingly on Page 2 Lines 20-21.

-I suggest to add between lines 33 and 36 a list of other multi-domain NRs solved using other biophysical techniques (SAXS, SANS) so that it is also easier to understand where the comments in the Results section – lines 93-1114 refer to. There are two references that should be also added in this manuscript, Rastinejad et al. TiBS (2015) and Moras D et al. in TiBS (2015).

Multi-domain NR structures solved by other techniques have been added on Pages 2-3. These two references suggested by the reviewer have also now been added to the manuscript.

-On page 3 line 60, it appears mentioned for the first time the technique hydrogen-deuterium exchange mass spectrometry, which the authors mention later on as either H/D-ex MS, H/D-ex or HDX. I suggest they use only one of these ways during the entire manuscript.

The abbreviations have been unified as H/D-ex MS throughout the manuscript.

-On page 3 lines 60-61 the authors refer to the fact that HDX can aid in determining whether there is allosteric signal transmission across the heterodimer to distal domains. I would like the authors to guide here much more the reader as to what do they mean by “distal domains”, to the ones belonging to one of the polypeptides or communication across the polypeptide chains.

By distal domains, we were referring to separate domains (ie. LBD versus DBD). These can be in one or other polypeptide, when detected by HDX.

-On page 4 line 71 please refer to the technique here used to measure the binding affinities to the DNA half-sites.

Fluorescence polarization (wherein we labelled the DNA duplex with a fluorophore) was the method we used for measuring the DNA-binding affinities. This information has been added on Page 7 Lines 19-21.

-On page 4 line 78, what NR box of SRC-2 have they used for the complex formation?

LXXLL synthetic peptides derived from SRC-2's second (middle) NR box were used for the crystallization as noted on Page 5 Line 1.

-On page 4 line 78, the search model used for molecular replacement could be already mentioned here.

The information about these search models (PDB codes: 1DSZ and 1XDK) has been added on Page 5 Line 3.

- On page 4 line 84, I would suggest a more descriptive way to explain how the two domains of RXRa are spatially displaced to each other.

A better description about the positions of RXR DBD and LBD has been included on Page 5 Lines 7-9.

-On pages 4 and 5 lines 84-85 and also in Figure 1c. Include the amino acid numbers that belong to the hinge domains that either have very poor or non-existent electron densities connecting the RXRa domains. This information can be added in the figure 1 legend so that the readers will understand what the dashed lines mean in the figure.

The legends of old Fig. 1c (now as Fig. 1b) have been updated with the information about the disordered residues as kindly suggested by the reviewer.

- On page 5 line 94 the authors refer to helix 12, it will be appreciated if this helix is labeled in some of the figures.

The helix-12 is now labeled in Fig. 1.

- Figure 1a, additional remarks need to be added to the schematic cartoon depicting RARb, the authors mention residue 414 as the end of the construct that has crystallized. They should add the number 448 at the end of this cartoon to show that this polypeptide misses C-terminal residues.

Figure 1a has been updated accordingly.

-Figure 1d, the DBD-LBD interface participating residues are shown as stick models. Add more level of detail to indicate the kind of contacts, hydrogen-bonds and the buried surface of each domain.

Figure 1d and its legends now indicate the hydrogen bonds at this DBD-LBD interface. More descriptions including the buried surface area (BSA) values have also been added in the text on Page 6 Lines 17-18.

-For non-experts on the H/D-ex MS patterns observed, I would suggest on page 6 lines 127-129 to clarify what are we expected to see on figure 2.

A brief explanation about how to understand H/D-ex MS data regarding to Fig. 2 has been added on Page 6 Line 23 and Page 7 Lines 1-3.

-On page 6 line 127, the authors state that “clear changes within the H/D-ex MS patterns” are observed. Please clarify what kind of changes, the details of what they refer to.

The text has been slightly modified to clarify this point on Page 7 Lines 4-7.

-On page 7 line 152 onwards, a series of mutations at the DBD-LBD junction have been studied. What was the rationale behind the chosen point-mutations, were they only a biochemical exercise or also some of these mutations have been associated to pathology?

These point mutations are not associated with any particular pathology. Instead, the choice of the mutations in this study was made purely based on their locations between the LBD and DBD within the context of their physical connections. These mutational choices allow us to decipher the importance of the DBD-LBD connections in biochemical and functional assays.

-On page 8 line 174, the authors state that significant variations have been clearly observed in the relative locations of the DBDs and the LBDs. Have the authors calculated r.m.s.d? Please give more detail here for structural biologists.

In examining the positions of DBDs and LBDs within different NR dimers, one can clearly see substantial differences from one NR to another NR (Figure 4). We aligned the DR1 elements to be identical to the viewers in these pictures. The rmsd values are difficult to calculate, and in our view not very helpful assessments, to further clarify such substantial differences. The text on Page 9 Line 23 and Page 10 Lines 1-2 has been modified to also direct the readers to Supplementary Fig. 5, where we also have superimposed the different NR complexes, to further highlight their dramatic differences with regard to their domain positioning.

-On page 9 line 178, the structures solved using SAXS are different than the ones solved using X-ray crystallography. The authors use the term “inconsistent”. Would it be possible that the protein in solution

and the protein conformation that crystallized present major differences, which can be attributed solely to the fact that different conformational minima or preferred conformations only occur when the protein is in solution, and when in the form of a crystal the neighboring molecules (crystal packing) contribute to these major differences. Would it be possible that this NR conformational variability makes both models possible according to the methods used and be both likely to exist in different cellular conditions? Does the model obtained by one biophysical technique necessarily invalidate the model obtained using the other technique, even if we leave aside the resolution obtained by the most powerful technique?

Some of these issues and questions are already discussed at length in the references cited, particularly refs 8-10). We did add additional points (the very end of our results sections) to address why certain types of biophysical studies may not be sufficient or robust enough to use as the sole basis for generating reliable “structural models” on par with crystallographic models.

-On page 13 line 283. How many independent experiments and measurements were done, please mention it here.

All luciferase experiments were repeated at least twice. This has also been mentioned on Page 14 Lines 13-14.

REVIEWERS' COMMENTS:

Reviewer #2 (Remarks to the Author):

The work presented here adds knowledge to what is known about nuclear receptors and their structure and dynamics. The authors have addressed all concerns Satisfactorily and I do not have any additional comments.

Reviewer #3 (Remarks to the Author):

The authors have carefully revised the manuscript to integrate the suggestions and questions I made. I believe this is an important contribution and deserves publication. In my opinion, the manuscript is ready now for publication.